# Design of New Concept of Knitted Hernia Implant

**DOI:** 10.3390/ma15072671

**Published:** 2022-04-05

**Authors:** Bogusława Żywicka, Marcin Henryk Struszczyk, Danuta Paluch, Krzysztof Kostanek, Izabella Krucińska, Krzysztof Kowalski, Kazimierz Kopias, Zbigniew Rybak, Maria Szymonowicz, Agnieszka Gutowska, Paweł Kubiak

**Affiliations:** 1Pre-clinical Research Center, Wroclaw Medical University, Pasteura 1, 50-367 Wroclaw, Poland; danuta.paluch@umw.edu.pl (D.P.); zbigniew.rybak@umw.edu.pl (Z.R.); maria.szymonowicz@umw.edu.pl (M.S.); 2Institute of Security Technologies “MORATEX”, Curie-Sklodowskiej 3, 90-505 Lodz, Poland; mstruszczyk@moratex.eu (M.H.S.); agutowska@moratex.eu (A.G.); pkubiak@moratex.eu (P.K.); 3Łukasiewicz Research Network—Textile Research Institute, Brzezińska 5/15, 92-103 Lodz, Poland; krzysztof.kostanek@iw.lukasiewicz.gov.pl; 4Department of Material and Commodity Sciences and Textile Metrology, Faculty of Material Technologies and Textile Design, Technological University of Lodz, Zeromskiego 116, 90-924 Lodz, Poland; ikrucinska@p.lodz.pl; 5Department of Knitting Technology, Faculty of Material Technologies and Textile Design, Technological University of Lodz, Zeromskiego 116, 90-924 Lodz, Poland; kkowalski@p.lodz.pl (K.K.); kkopias@p.lodz.pl (K.K.)

**Keywords:** PACVD, hernia knitted implants, post-implantation effect, viscera adhesion prevention, irritation

## Abstract

A knitted implant, unilaterally modified with plasma-assisted chemical-vapor deposition (PACVD), and with a nano-layer of fluorine derivative supplementation, for reducing the risk of complications related to adhesions, and the formation of a thick postoperative scar was prepared. The biological evaluation of designed or modified medical devices is the main aspect of preclinical research. If such studies use a medical device with prolonged contact with connective tissue (more than 30 days), biocompatibility studies require a safety assessment in terms of toxicity in vitro and in vivo, allergenicity, irritation, and cancerogenicity, reproductive and developmental toxicity. The ultimate aspect of biological evaluation is biofunctionality, and evaluation of the local tissue response after implantation, resulting in the determination of all aspects of local biocompatibility with the implemented synthetic material. The implantation of PACVD-modified materials in muscle allows us to estimate the local irritation effect on the connective tissue, determining the risk of scar formation, whereas implantation of the above-mentioned knitted fabric into the abdominal wall, assists with evaluating the risk of fistula formation—the main post-surgical complications. The research aimed to evaluate the local reaction of the soft tissues after the implantation of the knitted implants modified with PACVD of the fluoropolymer in the nanostuctural form. The local effect that occurred during the implantation of the designed implants was quantitatively and qualitatively evaluated when PACVD unmodified (reference), and modified medical devices were implanted in the abdominal cavity (intra-abdominal position) for 12 or into the muscles for 56 weeks. The comparative semi-quantitative histological assessment included the severity of inflammatory cells (multinucleated cells, lymphocytes, plasma cells, macrophages, giant cells) and the tissue response (necrosis, neovascularization, fibrosis, and fat infiltration) on a five-point scale. The knitted implants modified by PACVD did not indicate cumulative tissue response when they were implanted in the muscle and intra-abdominally with direct contact with the viscera. They reduced local tissue reaction (score −2.71 after 56 weeks of the implantation) and internal organ adhesion (irritation score −2.01 and adhesion susceptibility −0.3 after 12 weeks of the implantation) compared with the reference (unmodified by PACVD) knitted implant, which had an identical structure and was made of the same source.

## 1. Introduction

Knitted structures that are characterized by relatively high performance (high strength and relatively quick integration with connective tissue) are clinically applied to reconstruct connective tissue, including hernia, gynecological and urological reconstructions. Hernia meshes are used for the reconstruction of muscle fascia and recovery of the anatomical localization of the viscera.

Hernia repair, especially by laparoscopy, is the most often performed procedure in general surgery [1,2,3,4,5,6].

The open techniques for hernia reconstruction are used in several post-surgical complications but have a relatively high cost because of the relatively long period of hospitalization [7]. With the previous consideration, less-invasive techniques were developed over the last 40 years. The main problem in laparoscopic hernia surgery is the increased risk of viscera adhesion to the synthetic materials if the hernia implant is placed in the sublay position.

Hernia mesh can be designed using synthetic polymer, including Polypropylene (PP) [3,8,9,10,11,12,13,14], polytetrafluoroethylene (PTFE) [8,11,12], polyvinylidene fluoride (PVDF) [12,15], polylactide (PLA) [8,11,13,14], polycaprolactone (PCL) [8,11,14], polyglycolide (PGA) [8,11,12,14], or biological-tissue-derived materials (in form of membrane, fibers, films; absorbable or semi-absorbable) [8,11,14,15] as well as a combination of the above—results in composite materials [11,12,14].

Implants available on the market are characterized by several disadvantages, such as:the random arrangement of monofilament and the difference in the kind of the polymer in the hybrid knitted implants, resulting in a reduction in the assumed performance and safety and a significant rise in the risk of internal organ/implant adhesion if the hernia mesh is implanted using laparoscopic surgery [15,16,17];a significantly high surface density (e.g., for hernia meshes made entirely of PVDF monofilament, surface density >> 120 g/m^2^ relative to the surface density of knitted, polypropylene implants < 80 g/m^2^), resulting in a lack of biomimetics and risk of thick postoperative scar formation (reduction in patient comfort, complications long after surgery, breathing problems, pain, etc.) [15,17,18,19];a prolonged period of soft tissue over-growth in the case of implants made entirely of synthetic materials and at the same time a decrease in the strength of the created implant/tissue composition [15,18,19,20].

The ideal implant for hernia surgery should be used for the repair of fascial defects and allow for the growth of connective tissue, providing the lowest possibility of adhesion to viscera (if localized in direct contact), generating minimal systemic and local reactions [21].

Taking into account the aspects above, the design of a knitted implant, unilaterally modified with plasma-assisted chemical-vapor deposition (PACVD), would reduce the risk of complications related to adhesions and the formation of a thick postoperative scar. Moreover, an appropriate implant design achieved by unilateral insertion of the three-dimensional structure (unilateral looping of the knitted structure on the surface in direct contact with the connective tissue) will increase the contact area with the tissue and accelerate the implant integration into the patient’s tissue from the muscle fascia side.

To the best of our knowledge, there has been no research on the use of PACVD for deposition of the layer of fluoroorganic compounds on the surface of knitted polypropylene implants. Grendyorov, et al. [22] performed a deposition of SiOx-doped amorphous hydrocarbon (a-C:H:SiOx) coating onto the titanium (Ti-6Al-4V) alloy substrate using PACVD. The obtained materials supported surface endothelization of a titanium implant applied in direct blood contact and resulted in the absence of cytotoxicity. Silicon- and oxygen-incorporated hydrogenated amorphous carbon layers deposited onto the surface of the polypropylene solid material by PACVD showed low thrombogenicity and cytotoxicity in test in vitro, as well as reduced platelet adhesion and aggregation [23]. In [24] process of Ta and Ta/TaN deposition on AISI 316L stainless steel by PACVD technique was performed. The designed modifications yieldedimproved corrosion resistance and biocompatibility confirmed by the cytotoxicity test. Yin, et. al. realized that research on a-C:H (a-C:H:SiOx) layer deposition on silicon wafers, 304 stainless sheets of steel and M2 HSS sheets resulted in an increase in the friction coefficient and an improvement in anti-corrosion performance [25].

The main attribute of the preclinical evaluation of the designed or significantly modified medical devices is biocompatibility, which is helpful in determining safety and performance. The risk analysis, which was prepared according to the guidelines of the EN ISO 14791 and EN ISO 10993 Standards, provides the research programme for evaluating the biocompatibility of medical devices [4,26]. Moreover, the PN-EN ISO 10993-1:2010 Standard contains the guide for selecting the most appropriate test methodology to evaluate the safety and some aspects of the performance of the designed medical devices. It decreases the risk of adverse effects during both clinical studies and standard clinical use.

The best test methodologies that allow valid evaluation of the functionalities of medical devices must be designed because of the wide range of medical device functionalities, a wide scope of clinical applications, and the relatively quick evolution of new medical technologies.

The previous works were performed to optimize the physical parameters of newly designed implants [27,28], study the effect of PACVD-modification on physical behavior [29], the chemical characterization of leachable profile [30], the evaluation of the PACVD-modification effect after accelerated tests [31], the biomechanical evaluation in the simulated environment [32] and finally, the assessment of structural properties after explanation [33]. The research carried out with the PACVD-modified knitted implants indicated the presence of nano-pores with the average surface size of 4.4. nm, and average pore size of 68.7 nm (determined by mercury porosimetry) on the surface of the filaments. The above-mentioned surface behavior claims that there is a low risk of microbial accumulation in pores not less than their diameter—the main risk of postoperative complication (infection) resulted from the synthetic materials. 

The EN ISO 10993-1 Standard provides only general indications for properly selecting the required biocompatibility test to evaluate the hernia implants. The basic tests, which should be performed to estimate the potential risk of the adverse events of the designed medical devices, cover cytotoxicity, allergenicity, irritation, genotoxicity, systemic toxicity—chronic toxicity, and local effect after implantation. Moreover, the carcinogenicity, reproductive and developmental toxicity should be considered to evaluate the safety of these devices based on the results of the risk analyses. The adhesion effect of internal organs to the hernia implants is not estimated using the previously mentioned standard.

The research aimed to evaluate the local effect after implanting knitted implants that were modified using PACVD of the fluoropolymer in the nanostructural form. The tested implants surface of the tested implant was marked using a stitch of surface loops.

During the study, the methodology derived from the EN ISO 10993-6 Standard was enhanced by the new anatomical placement of the test in the abdominal cavity. Additionally, based on the prepared concurrent analysis, the viscera adhesion risk estimation method [34] was improved using two semi-quantitative criteria: the surface area of the adhesion and the adhesion mode.

The local effect that occurred during the implantation of the designed implants was qualitatively and quantitatively evaluated when PACVD unmodified (reference) and modified medical devices were implanted in the abdominal cavity (intra-abdominal position) or within the muscles.

The evaluation methodology proposed in this study will estimate the viscera adhesion risk for hernia implants. The novelty of the research is focused on the modification of the knitted implants by PACVD for:the reduction of the risk of the implant to the viscera (if direct contact of the implant is necessary);acceleration of the connective tissue in-growth due to the enhancement of the implant surface promoting the fixation of the implant.

## 2. Materials and Methods

### 2.1. Knitted Implants

The prototypes of the hernia implants were knitted using polypropylene monofilament yarn (Sider ARC S.p.A, Cornaredo, Italy) with a diameter of 0.08 mm (46 dtex) made of polypropylene class VI polymer acc. US Pharmacopeia. The process of knitting was performed using the variants of three-dimensional (3-D) textile structures manufactured by the knitting technique HKS3M knitting machine (Karl Mayer, Obertshausen, Germany). The process of the design is described in [17]. The structure of the designed prototype is presented in Figure 1.

The one-side process of the PACVD modification of the designed hernia implant with the presence of tetradecafluorohexane (Fluka, Buchs, Germany) is described in [27,36]. The physiochemical properties of the reference and the studied prototype (after PACVD modification and related reference) are presented in Table 1.

The PACVD modified (tested mesh) and unmodified knitted implants (control mesh) were applied during the study. Both knitted structures were placed in a clean room in a packaging system (OPM, Toruń, Poland) and then sterilized using steam at a temperature of 121 °C for 30 min. at the TZMO SA sterilization plant, Lodz, Poland (Figure 2).

### 2.2. Methods

The research into the local response of the muscle tissue, and the intra-abdominal implantation of the tested samples and the reference mesh were studied according to the guidelines of the PN-EN ISO 10993-6:2007 Standard. For the tests performed, the approval of The Ist Local Ethical Committee for Animal Experiments in Wrocław (Wrocław, Poland) was obtained—No. 45/2008. Rabbits were anesthetized intramuscularly with xylazine at a dose of 5 mg/kg body weight (Sedazin, Biovet, Puławy, Poland) and Ketamine at a dose of 30 mg/kg body weight (Bioketane, Vetaquinol, Gorzów Wielkopolski, Poland). The state of basic sleep and full analgesia was obtained by the animals 10–15 min after the preparate injection and lasted for 60–80 min.

#### 2.2.1. Local Effect after Implantation in Muscles

The research included implantation, a clinical examination, necropsy, macroscopic and histological evaluation of the response of the muscle tissue to the presence of the tested mesh in comparison to the reference sample in periods 1, 3, 9, 12, 25, or 56 weeks. During the study, the risk level was rated in terms of the potential use of the developed product regarding a local biological incompatibility.

1 cm in diameter disc-shaped materials (PACVD-modified and reference) were implanted in pockets in the vertebral muscles of New Zealand White (NZW) rabbits. On both sides of the spine, 3 pockets in the muscles were made in which the tested materials (left side) and reference (opposite side) were implanted (Figure 3).

During the clinical evaluation, the condition of the animals was verified based on their behavior, skin condition, the appearance of the natural body orifices, and the status of the surgical wounds.

During an autopsy the condition of the natural orifices of the animal, the changes in muscles and skeleton, the appearance of the surgical scar, and the condition of the rabbit’s back muscle fascia after skin removal where the implant and reference were evaluated during each skin necropsy. Additionally, the appearance and arrangement of the internal cavity organs were macroscopically evaluated. Fragments of muscle tissue with implants (of both the tested material and the reference) were collected for microscopic examination.

In the frame of research, histological microscopic assessments, both qualitative and semi-quantitative, were performed. Additionally, the dynamics of the healing process were assessed.

The semi-quantitative evaluation of the histopathological response for the knitted implants and the estimation of the tissue response was performed based on the criteria described in the Annex E; Tables E1—E2 of PN-EN ISO 10993-6:2007 Standard (2 in the scope of the presence of the inflammatory cells (polymorphonuclear cells, lymphocytes, plasma cells, macrophages, and giant cells) and the resulted tissue response (necrosis, neovascularization, fibrosis, and fatty infiltrates).

Cellular response for the implant was graded on a scale: 0—(0 cells phfa), 1—(1–5/phf), 2—(5–10/phf), 3—heavy infiltrate, 4—packed phf = per high powered (400×) field.

Similarly, a 5-point scale ranging from minimal to extensive lesions was used in the histological evaluation of the tissue response. Averaged data were summed for each study period, and the difference between the mesh and control data provided the basis for assessing the general tissue response.

Table 2 shows the evaluation score for the estimation of the general response of implants tested.

#### 2.2.2. Local Effect after Implantation in the Abdominal Wall

Rectangular samples (25 mm × 40 mm) were implanted in aseptic conditions in NZW rabbits by fixing the mesh on the inner side of the abdominal wall, with the evaluated sample on one side and the reference on the opposite side (Figure 4). To avoid the aggravation of the tissue reaction, the stabilizing stitches were left outside.

After implantation, the animals were subjected to post-surgical observation (overall appearance and behavior, skin condition, appearance of the natural orifices, and surgical wounds) for the duration of the study. During the necropsies after 2, 4, or 12 weeks, the skin, natural body orifices, changes in the animal’s muscles and skeletal system, and the state of the surgical wounds were assessed. Additionally, after skin removal, the condition of the abdominal wall surface, where the samples were implanted, was evaluated. Moreover, a macroscopic evaluation of the appearance and arrangement of the following organs in the peritoneal cavity was performed in the following order: stomach, intestine, liver, pancreas, spleen, lymph nodes, kidney, bladder, and internal genitals.

As part of the study, the risk was estimated for the following:-adhesion of the tested sample to the viscera in direct contact of the surface with a nanolayer of PACVD originated fluoropolymer or non-modified reference with them as a result of intraperitoneal implantation;-local, potential biological incompatibility when the developed medical device is used as a textile implant for hernia reconstruction.

The viscera adhesion to the implant was estimated based on the semi-quantitative macroscopic analysis (Table 3).

The susceptibility for implant adhesion to the viscera was estimated based on (1) the calculation of the differences between the score for the adhesion area (Table 3; criterion [x_2_]) and (2) the calculation of the weighted average, according to the following equations:

(1) differences between the score for the adhesion area:(1)$$S_{a1}=\frac{\sum_{i=1}^{n}{\left[x_{2m}\right]}_{i}}{n}-\frac{\sum_{i=1}^{n}{\left[x_{2c}\right]}_{i}}{n}$$
where:

*S_a_*_1_—susceptibility for the implant adhesion to the calculated viscera considering the adhesion area (Table 3; criterion [x_2_]);

[*x*_2*m*_]—individual adhesion area determined for the evaluated sample;

[*x*_2*c*_]—individual adhesion area determined for reference;

*n*—number of explanted samples;

*S_a_*_1_ < 0: the evaluated sample shows lower adhesion compared with the reference (better biocompatibility).

(2) differences among the scores of the weighted average, as defined in Table 3. Criteria (adhesion quality, adhesion area, and mode of the implant covering): (2)$$S_{a2}=\;\frac{\sum_{i=1}^{n}w_{i}\times y_{i}}{\sum_{i=1}^{n}w_{i}}-\frac{\sum_{i=1}^{n}w_{i}\times z_{i}}{\sum_{i=1}^{n}w_{i}}$$
where:

*S_a2_*—susceptibility for the implant adhesion to the viscera based on the following criteria: adhesion quality, adhesion area, and mode of the implant covering (Table 3; criteria [x_1_], [x_2_], and [x_3_]);

*w_i_*—importance for the estimated criterion (Table 3);

*y_i_*—average score of the estimated criterion for the evaluated sample;

*z_i_*—average score of the estimated criterion for the reference;

*n*—number of criteria;

*S_a_*_2_ < 0: the evaluated sample shows lower adhesion compared with the reference (better biocompatibility).

The overall susceptibility of implant adhesion to the viscera was calculated according to the following equations:(3)$$OS_{a1}=\overset{n}{\underset{i=1}{\sum }}{\left(S_{a1}\right)}_{i}$$
where:

*OS_a_*_1_—overall susceptibility of implant adhesion to the viscera, calculated based on criterion [x_2_];

(*S_a_*_1_)*_i_*—susceptibility of implant adhesion to the viscera, calculated for the individual explanation period based on criterion [x_2_].
(4)$$OS_{a2}=\overset{n}{\underset{i=1}{\sum }}{\left(S_{a2}\right)}_{i}$$
where:

*OS_a_*_2_—overall susceptibility of implant adhesion to the viscera, calculated based on criteria [x_1_], [x_2_], and [x_3_];

(*S_a_*_2_)*_i_*—susceptibility for the implant adhesion to the viscera, calculated for the individual explanation period based on criteria [x_1_], [x_2_], and [x_3_].

Additionally, the semi-qualitative evaluation of the histopathological response for the knitted implants, and the estimation of the tissue response were made based on the requirements described in Annex E; Table E.1–E.2 of PN-EN ISO 10993-6:2007. The evaluation score for the estimation of the general response of the tested implants was similar as that described in Table 2.

## 3. Results and Discussion

### 3.1. Study of the Local Effect after Implanting the Knitted Implants within the Muscles

In the ATR—FTIR spectrum of PACVD-modified implants, the presence of absorption bands at a wavenumber of 1339 cm^−1^ associated with the presence of a modified surface (C–F bond) at the stage of PACVD was found. The process of sterilization did not affect the presence of the above-mentioned absorption band. There was a significant increase in the intensity of the absorption bands at λ = 2870 cm^−1^, λ = 2920 cm^−1^ (with the absorption band shift to a higher wavenumber at λ = 2917 cm^−1^), and λ = 2950 cm^−1^ as compared with the unmodified implant ATR-FTIR spectrum [38]. A similar phenomenon was found where the other types of the polymer (*p*-aramid, UHMWPE) have been undergoing the PACVD with the presence of the tetradecafluorohexane. Moreover, two broad absorption peaks were detected at wavenumber 1240 ÷ 1205 cm^−1^ [39].

A quantitative EDS microanalysis of the fluorine content on the implant surface directly after the PACVD modification, showed a value of 5.22%. This value was slightly reduced after the steam sterilization process to 4.98%. The degree of crystallinity of the sterile implant determined by the WAXS technique was 58.8% [38].

The preliminary study-verification of the degree of implant adhesion was performed taking into account the mechanical test (adhesion strength), SEM observation of the explanted implants, and changes in the crystallinity index as well as FTIR verification of the explanted implant after primary purification from the tissue [33]. The conducted tests (semi-quantitative evaluation of adhesion according to the developed evaluation criteria) showed that the PACVD-modified knitted implants adhered on a significantly smaller surface than in the case of the control sampling. A similar phenomenon was also validated in the quantitative assessment of adhesion exvivo [33]. 

The effect of implantation (2, 4, or 12 weeks) on the structural properties of the developed implants with respect to an unmodified sample, the qualitative adhesion degree of internal organs to the modified surface (by SEM), as well as the overgrowth degree by the connective tissue of the unmodified surface with loops was also validated [33]. 

The statistically insignificant changes in crystallinity index during the intra-peritoneal implantation of the PACVD up to 12 weeks were observed. The ATR-FTIR tests confirmed the presence of the fluoropolymer layers (presence of absorption band at λ = 1339 cm^−1^ corresponding to C–F bond) even after 12 weeks of the implantation [33].

These studies ex vivo confirmed the better integration of the PACVD-modified implant into connective tissue, the stability of the implemented fluoropolymer layer, and the reduction in the risk of adhesions with internal organs.

During the clinical observations of the treated animals in the macroscopic study, no skin lesions were found. The skeletal, muscular system, and natural body orifices showed no pathological lesions.

During the necropsies at 1, 3, 9, 12, 26, or 56 weeks after implantation, the postoperative wounds of the skin and the dorsal muscles, where the reference and tested samples were implanted, were properly healed by primary intention. In the macroscopic study, the back muscles appeared to be correct in drawing and color. The tested sample and reference were inherent and strongly tied to the muscle tissue with normal, unaltered appearance) (Figure 5).

In histological studies, in the early period, around the monofilaments of implants, bands of loose connective tissue were found in both the test and the control groups; these bands surrounded the individual filaments of the implant and separated their structure from the muscle tissue (Figure 6).

During the soft inflammation process, in the third week after the sample implantation, the presence of vascularized, fibrous connective tissue was observed. In the immediate vicinity of the mesh fibers, loose connective tissue with single giant cells was present. (Figure 7).

In further stages of healing, the structure of the knitted implants was interlaced with fibrous connective tissue with fatty infiltrates at muscle side, whereas loose connective tissue with single polynuclear macrophages in the immediate vicinity of the implant structures was observed (Figure 8).

The semi-quantitative, histological evaluation of the local reaction of the muscular tissue showed that the tested sample did not exhibit any irritating impact (according to the criteria of PN-EN ISO 10993-6:2006 Standard) as compared with the reference. Moreover, prolonging the implantation of the test sample reduced the local response of muscle tissue compared with the control sample.

The results achieved from the semi-quantitative evaluation of the muscle tissue response for the knitted implants (tested sample and reference) are shown in Table 4.

The first part of the study showed that the mono-layer of fluoropolymer placed onto one side of the PACVD-modified knitted implants to reduce the local muscle reaction in the long term of the implantation (longer than 26 weeks) confirmed a significant decrease in the risk of massive connective tissue formation yielding in the stiffening of the abdominal muscle. The introduction onto one side of the prototype hernia mesh fluoropolymer layer, improved biocompatibility by the reduction of the local irritation reaction as compared with the non-modified reference.

Moreover, no infection was observed confirming the positive effect of reduction of the pore sizes in the structure of the knitted implants made of the PACVD-modified monofilament.

### 3.2. Study of the Local Effect after Intra-abdominal Implantation of the Knitted Implants

In the post-surgery period, in both the reference and PACVD modified sample group, the animals behaved normally. Throughout the research period, all of the rabbits were clinically healthy. During the necropsies, which occurred after 2, 4, or 12 weeks, no skin lesions, no changes in the muscular, skeletal system, or natural orifices were found. The post-surgery wounds of the skin or the abdominal wall were properly healed by the primary intention at the site of implantation of either the tested sample or the reference.

During the macroscopic examination, no changes were found in the appearance or arrangement of the peritoneal cavity organs (stomach, intestines, liver, pancreas, spleen, lymph nodes, kidney, bladder, and internal genitals). All of the organs in the chest and peritoneal cavity were properly arranged with preserved anatomical shape, color, and size and without macroscopically visible lesions. In the macroscopic view of the internal organs (stomach, intestines, liver, pancreas, spleen, lymph nodes, kidneys, bladder, and internal genitals), no pathological changes were found.

Samples of the tested and the referenced groups, which were covered with a whitish-pink bag, were located on the inner side of the abdominal wall of the natural color and shape, and they were not relocated. In individual cases, the samples were slightly bent. The internal organs or intestinal mesentery adhere to the implant surface to various extents.

Treatment with the proposed device contributes to reducing local tissue reaction. 

The susceptibility for implant adhesion to the viscera was estimated by calculating (1) the differences between the score for the adhesion area; (*S_a_*_1_) and (2) the weighted average of the score for all of the criteria defined in Table 3; (*S_a_*_2_).

Table 5, Table 6 and Table 7 contain the results of the intra-abdominal implantation of the PACVD modified implant in relation to the reference (unmodified by PACVD) after 2, 4, or 12 weeks.

The susceptibility for the adhesion [*S_a_*_1_], which was calculated based only on the score for the adhesion area, yielded the following results (Table 5, Table 6 and Table 7; considering the criterion Adhesion area):-2 weeks after the intra-abdominal implantation—0 points;-4 weeks after the intra-abdominal implantation—(−1.4) points;-12 weeks after the intra-abdominal implantation—(−0.6) points.

*OS_a_*_1_ (−2.0 points), which is the sum of the individual *S_a_*_1_ that was calculated for the individual ex-plantation period, indicated the reduction in risk of viscera adhesion to the implants, considering the area of the adhesion. Two other criteria were estimated: adhesion quality [28] and mode of the implant covering, both of which significantly improved the method for the adhesion-risk calculation.

The semi-quantitative, macroscopic assessment of the adhesion rate of PACVD-modified prototype (the susceptibility for implant adhesion to the viscera; *S_a_*_2_), which was the first round of necropsy on the 14th day after the implantation, demonstrated no difference between the tested material and reference. The median level of the weighted average for scoring was over 2 points (2.0 points for the reference and 2.2 points for the tested sample—PACVD-modified knitted implant), and *S_a_*_2_ was 0.20 points. After 4 weeks, *S_a_*_2_ of the tested implant was reduced to −0.80 points because the tendency in weight average of the scoring of the tested implant for three criteria was reduced. This observed trend continued in the third round of necropsy (12 weeks after implantation), with the average score of the tested implant prototype and the referenced sample 1.5 points and 1.8 points, respectively, and *S_a_*_2_ was (−0.30) points. *OS_a_*_2_ (−0.9 points) yielded a similar conclusion due to the criterion of adhesion to area being the only one selected to calculate implant susceptibility to adhesion.

The results of both semi-quantitative evaluations of adhesion indicate that the prototype of knitted, PACVD surface-modified implants adhered to a smaller surface compared with the reference (knitted fabric characterized by identical structure and contact area, but without the PACVD-modification). 

The nanolayer of fluoropolymer originated from PACVD modification and localized on one side of the implant (directly to viscera) showed a significantly lower risk of internal organ adhesions. 

Samples of the abdominal walls were collected for histopathological examination, with samples of tested and reference material, and adjacent organs (small intestine, large intestine, mesentery). A microscopic examination in the early post-implantation period, 2 weeks after implantation, indicated the presence of a band of loose connective tissue surrounding the individual monofilaments of the knitted implant, separating its structure from the muscles and peritoneal cavity (Figure 9).

Four weeks after implantation, fibrous connective tissue is formed, and eosinophiles (e) accumulated in the implant structure, occurring in the reference. In the immediate vicinity of the monofilaments, the loose connective tissue with single polynuclear macrophages was evident. The adipose tissue of intestinal mesentery was adjacent to some places of the described tissue (Figure 10). 

At a further stage of the healing process, 12 weeks after the implantation, the implant monofilaments were surrounded and interlaced with fibrous connective tissue with the adipose tissue or the intestinal walls that were adjacent to it in some places from the side of the peritoneal cavity. In the immediate vicinity of monofilaments of both the tested and the referenced implants, there was loose connective tissue (Figure 11).

The results of the semi-quantitative histological evaluation of the cellular and tissue response for the knitted implants that were modified using the PACVD in reference to the unmodified implant having identical knitted structure implanted in sublay position are shown in Table 8. 

The above phenomenon confirms the significantly lower, local tissue response due to the fluoropolymer-containing nanolayer added onto one side of the surface of the knitted implants (even 4 weeks after implantation and in the longest term of the experiment), confirming the macroscopic observation of the adhesion effect to the viscera. Observed changes in behavior of the modified surface of the knitted implants by adding the nanolayer of the hydrophobic polymer (which increase the wetting angle from 113° ± 8° to 143° ± 12°) significantly alters the connective tissue, and promotes the adhesion of implant to viscera risk reduction when direct contact with them. Additionally, due to the lower, local irritation effect promoting scar formation, resulted in the stiffening of the abdominal structures.

The enhanced semi-quantitative histological assessment indicated that the PACVD-modified sample in each period of the test showed no irritation effect as compared with the reference (based on the criteria described in PN-EN ISO 10993-6:2006 Standard). It should be noted, that the reference induced a slight irritation in the 4th week after surgery as compared with the tested implant prototype.

The results of the semi-quantitative histological evaluation after 12 weeks of the implantation indicated a high difference in score between the tested sample and the reference (favoring the designed implant—PACVD-modified knitted mesh).

## 4. Conclusions

Considering the collected results, the modification of the knitted implant surface using the PACVD with fluoropolymer deposition in the form of nanolayer on one side surface prevented viscera adhesion to the implant during intra-abdominal implantation. Moreover, the designed knitted implants did not induce the cumulative tissue in-growth when implanted either in the muscle or intra-abdominally. This phenomenon indicates that the PACVD originated nanolayer of fluoropolymer onto one side of knitted implants could be optimal for the reconstruction of hernia without the risk of overmatched scar formation. Also, no observed infection confirms the safety of the designed implant due to the reduction in pore size to nanoscale. 

The developed, knitted PACVD-modified implants with fluoropolymer deposition affected the local reduction in the tissue reaction compared with the reference (an implant that was unmodified by PACVD with the identical structure and fabricated from the identical source).

The risk of viscera adhesion to the implant usually increases when the contact surface is larger. The implants fabricated using deposition of the nanolayer of the fluorine-containing polymer onto low-diameter yarns, and a more macroporous structure with large surface loops reduce this adhesion. The PACVD modification that involved the deposition of a fluoropolymer nanolayer onto one side of the implant surface, adhering directly to the internal organs, resulted in an additional reduction in the risk of adhesion. 

The adhesion risk evaluation should be enhanced in other aspects than adhesion quality. The adhesion area and adhesion mode of the viscera to the implant appear optimal for a more comprehensive estimation of the performance of the hernia implants that directly contact the viscera.

Multiple criteria, as previously described, allow the more effective estimation of the adhesion risk, particularly when low-adhering implants (due to the fluoropolymer deposition onto the implant surface) combined with low-diameter monofilaments are used. Moreover, using the justified method, the improved methodology to evaluate the adhesion risk (based on the semi-quantitative criteria that cover the macroscopic and microscopic evaluation) allows for the precise estimation of the risk of viscera adhesion to the implant.

## Figures and Tables

**Figure 1 materials-15-02671-f001:**
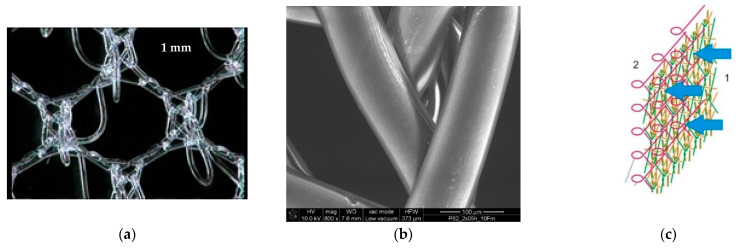
Optical microphotograph (**a**) and SEM microphotograph (mag, 800×) of the designed PACVD-modified prototype of the hernia implant (**b**) used in the study. The idea of PACVD-modification of the designed knitted implants: 1—the flat surface of the implant subjected to PACVD-modification; 2—surface not subjected to the modification with unfolding in the form of protruding loops (**c**) [35].

**Figure 2 materials-15-02671-f002:**
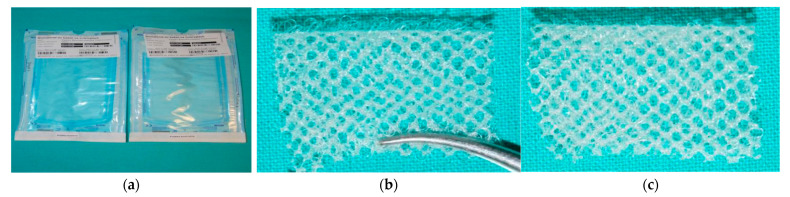
The knitted implants used to estimate the local effect after implantation: (**a**) materials for the test used; (**b**) macroscopic view of the surface with loops; (**c**) macroscopic view of the opposite surface without loops and PACVD-modified with the presence of the fluoropolymer.

**Figure 3 materials-15-02671-f003:**
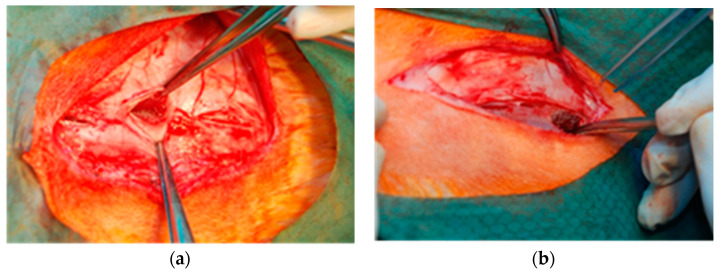
Intrasurgical photography: implantation of samples into pockets in the dorsal muscles of an NZW rabbit. The evaluated samples are localized on the left (**a**), and the reference on the right side of the spine (**b**).

**Figure 4 materials-15-02671-f004:**
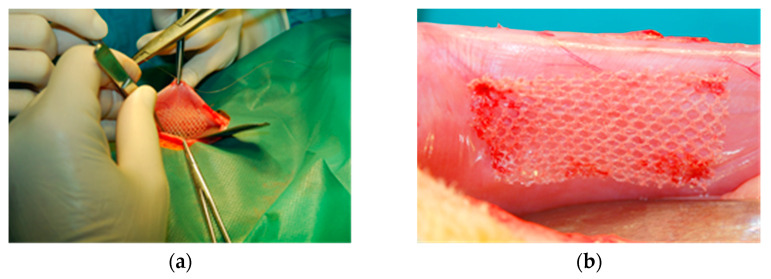
Intrasurgical photography: implantation of the tested implant into the abdominal wall (intraperitoneal position): (**a**)—the operative filed view; (**b**)—location of the implant.

**Figure 5 materials-15-02671-f005:**
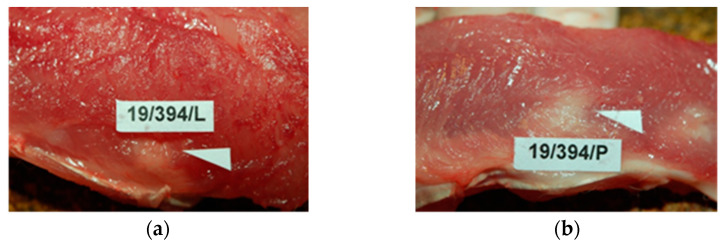
Macroscopic view of: (**a**)—the tested PACVD-modified sample and (**b**)—after 12 weeks of intramuscular implantation (implants are marked by arrows.

**Figure 6 materials-15-02671-f006:**
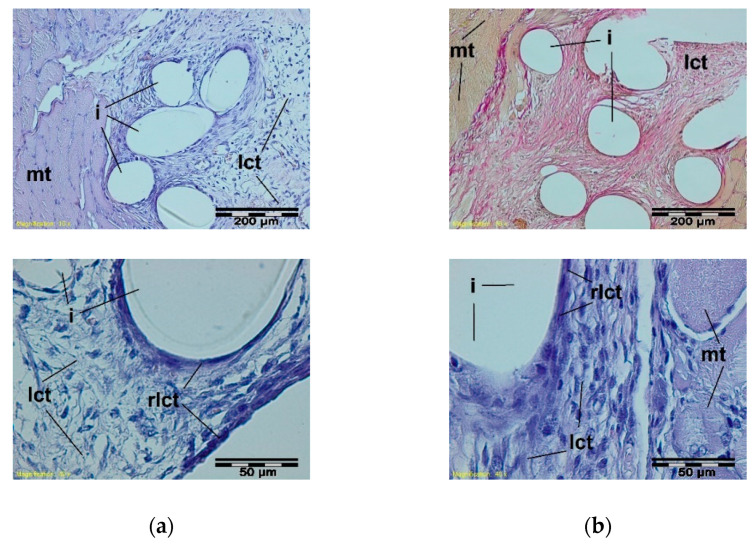
A microscopic view of the PACVD-modified sample (**a**) and the reference (**b**) 1 week after the intramuscular implantation. The monofilaments of knitted fabrics (i) surrounded by a loose connective tissue (lct) are separated from the muscle tissue (mt). A rich-in-cell, loose connective tissue (rlct) in the immediate vicinity of the monofilaments of the knitted structure is visible. Magnification 100×, Stain. HE and VG; Magnification 400×, Stain. HE.

**Figure 7 materials-15-02671-f007:**
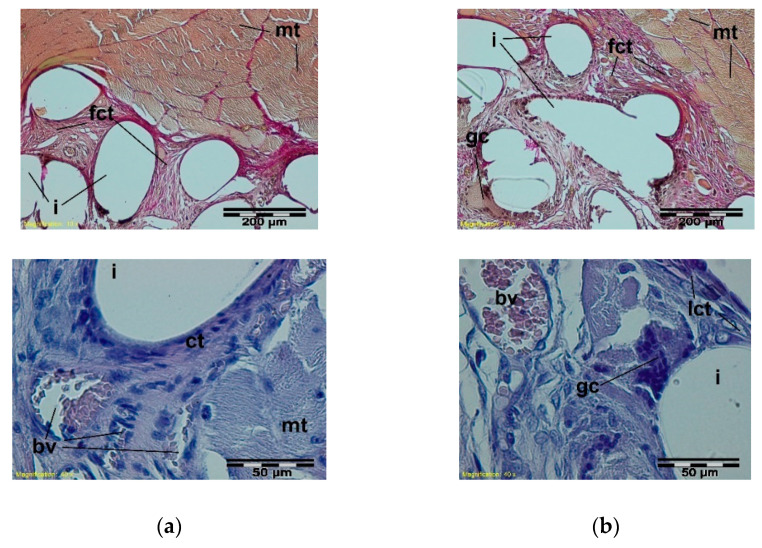
A microscopic view of the PACVD modified sample (**a**) and the reference (**b**) 3 weeks after the intramuscular implantation. In the muscles (mt), the implant monofilaments are surrounded by connective tissue (fct) with numerous blood vessels (bv). A loose connective tissue with single giant cells in the immediate vicinity of the knitted structure is visible. Magnification 100×, Stain. VG; Magnification 400×, Stain. HE.

**Figure 8 materials-15-02671-f008:**
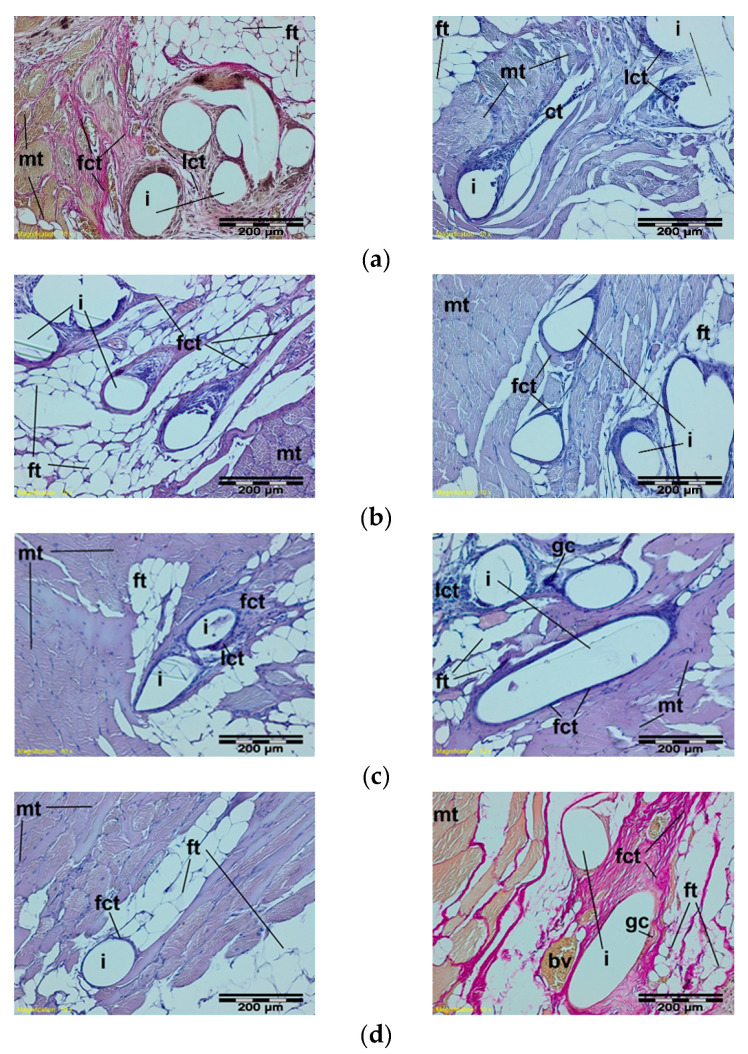
A microscopic view of the PACVD-modified sample (left side) and the reference (right side) after 9 (**a**), 12 (**b**), 26 (**c**), or 56 weeks (**d**) after the intramuscular implantation. In the muscle tissue (mt), the monofilaments of knitted implants (i) are surrounded by fibrous connective tissue (fct) with blood vessels (bv) and bands of fatty infiltrate (ft). Loose connective tissue (lct) with single polynuclear macrophages (gc) in the immediate vicinity of the implant structures (i) is visible. Magnification 100×, Stain. HE and VG.

**Figure 9 materials-15-02671-f009:**
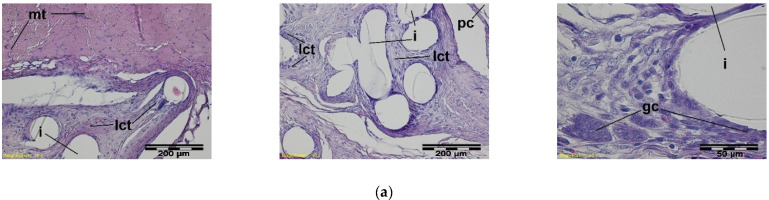
A microscopic view of the PACVD-modified implant (**a**) and reference (**b**) 2 weeks after the implantation in the abdominal wall of a NZW rabbit. Monofilaments of knitted structure (i) are surrounded by loose connective tissue; image from the muscle tissue (mt) side (on the left) and from the peritoneal cavity (pc) (in the middle). In the immediate vicinity of the fibers, polynuclear macrophages (gc) are formed (on the right). Magnification, 100× and 400×, Stain. HE.

**Figure 10 materials-15-02671-f010:**
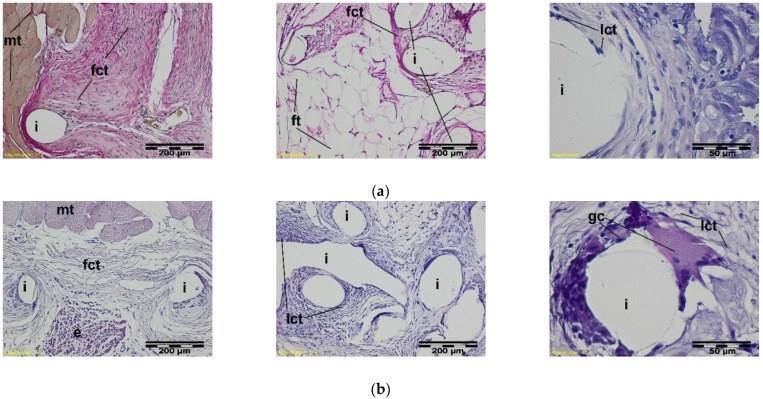
A microscopic view of the PACVD-modified implant (**a**) and reference (**b**) 4 weeks after the implantation in the abdominal wall of a NZW rabbit. The monofilaments of knitted structures (i) are surrounded by fibrous connective tissue (fct) from the side of the muscles (on the left) of the abdominal wall and peritoneal cavity (pc) (in the middle). On the right side, the loose connective tissue (lct) with polynuclear macrophages (gc) appears in the immediate vicinity of the monofilament. Magnification, 100×, Stain. VG and HE. Magnification 400×, Stain. HE.

**Figure 11 materials-15-02671-f011:**
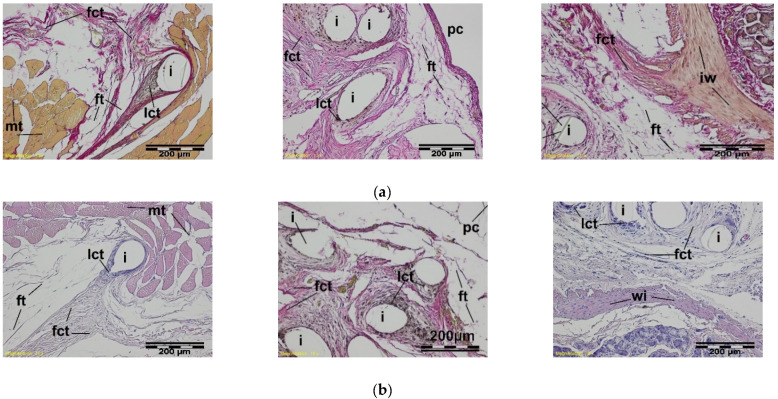
A microscopic view of the PACVD-modified implant (**a**) and reference (**b**) 12 weeks after the implantation in the abdominal wall of a NZW rabbit. Monofilaments of the knitted structure (i) are surrounded by connective tissue, which was fibrous (fct) from the muscles (mt) side and loose (lct) in the vicinity of the thread. On the right side, the adjacent intestinal wall (iw) is visible. Magnification, 100×, Stain. VG and HE.

**Table 1 materials-15-02671-t001:** Physiochemical properties of the reference and the studied prototype.

Sample	Test Method	Reference Sample	PACVD Modified Prototype (Evaluated Sample)
Surface density (g/m^2^)	PN-EN 12127:2000	45.9 ± 0.8	46.1 ± 1.5
Thickness (mm)	PN-EN ISO 5084:1999	1.08 ± 0.09	1.10 ± 0.01
Longitudinal tensile strength (N)	PN-EN ISO13934-1:2002	114 ± 10.1	89.0 ± 23.7
Vertical tensile strength (N)		103.0 ± 8.8	110.0 ± 12.8
Longitudinal elongation (%)		56.1 ± 12.4	76.4 ± 8.6
Vertical elongation (%)		72.0 ± 5.2	62.7 ± 8.7
Longitudinal initial elasticity modulus (MPa)		4.66	0.96
Vertical initial elasticity modulus (MPa)		2.66	2.4
Bursting Strength (N)	PN-EN ISO 12236:2007 ^1^	584.0 ± 53.8	564.0 ± 26.1
Suture pullout in the corner (N)	ISO 7198:1998 ^2^	20.7 ± 2.6	21.7 ± 3.8
Fluorine content onto the surface (%)	determined by SEM-EDS	-	4.98
Crystallinity index (%)	determined by WAXS	61.2	58.8
Wetting angle (^o^)	[37]	113 ± 8	143 ± 12

^1^ Using semispherical stamp. ^2^ Method adaptation.

**Table 2 materials-15-02671-t002:** The evaluation score for the general-response estimation of the tested implants.

Score (Difference in Score of the Tested Sample and the Reference)	Evaluation
0–2.9	No irritation
3.0–8.9	Slight irritation
9.0–15.0	Medium irritation
>15.1	Strong irritation

**Table 3 materials-15-02671-t003:** Semi-quantitative analysis of the viscera adhesion to the implant.

Criteria					
Adhesion Quality (x_1_)		Adhesion Area (x_2_)		Mode of the Implant Covering (x_3_)	
Absence	**0**	0–20% of implant area	**0**	Absence	**0**
The place of the adhesion can be lightly removed	**1**	20–45% of implant area	**1**	Point-to-point area	**1**
The place of the adhesion can be removed using additional strength (aggressive separation)	**2**	45–70% of implant area	**2**	Multipoint adhesion in several places onto the implant	**2**
The place of the adhesion can be removed using sharp tools (by cutting)	**3**	>70% of implant area	**3**	Full adhesion	**3**
Importance (w_1_)	**1**	Importance (w_2_)	**2**	Importance (w_3_)	**1**

**Table 4 materials-15-02671-t004:** The evaluation score for the general-response estimation of the tested implants.

Time of the Implantation (Week)	Score ^1^	Evaluation
1	2.60	No irritation
3	2.42	No irritation
9	−2.71	No irritation
12	2.99	No irritation
26	−2.05	No irritation
56	−2.71	No irritation

^1^ Score < 0: the evaluated sample shows a lower irritation reaction compared with the reference (better biocompatibility).

**Table 5 materials-15-02671-t005:** Semi-quantitative, macroscopic evaluation of the adhesion susceptibility of the PACVD-modified implants in relation to the reference (unmodified by PACVD) sample after 2 weeks of the intra-abdominal implantation.

	Reference			PACVD Modified Knitted Implant		
	Adhesion Quality	Adhesion Area	Mode of the Implant Covering	Adhesion Quality	Adhesion Area	Mode of the Implant Covering
	2	0	2	3	0	2
	2	3	2	2	3	2
	2	0	2	2	3	2
	2	3	2	2	0	2
	2	2	2	2	2	2
**Summary**	10	8	10	11	8	10
**Average**	2.0	1.6	2.0	2.2	1.6	2.0
** *S_a_* _1_ **	0					
**Importance**	1	2	1	1	2	1
**Weighted average**	1.8			2.0		
** *S_a_* _2_ **	0.20					

**Table 6 materials-15-02671-t006:** Semi-quantitative, macroscopic evaluation of the adhesion susceptibility of the PACVD-modified implants in relation to the reference (unmodified by PACVD) sample after 4 weeks of the intra-abdominal implantation.

	Reference			PACVD-Modified Knitted Implant		
	Adhesion Quality	Adhesion Area	Mode of the Implant Covering	Adhesion Quality	Adhesion Area	Mode of the Implant Covering
	2	1	2	1	1	2
	1	1	2	2	0	1
	2	3	3	2	1	3
	2	3	3	2	1	3
	2	3	3	2	1	2
**Summary**	9	11	13	9	4	11
**Average**	1.8	2.2	2.6	1.8	0.8	2.2
** *S_a_* _1_ **	−1.4					
**Importance**	1	2	1	1	2	1
**Weighted average**	2.2			−1.4		
** *S_a_* _2_ **	−0.8					

**Table 7 materials-15-02671-t007:** Semi-quantitative, macroscopic evaluation of the adhesion susceptibility of the PACVD-modified implants in relation to the reference (unmodified by PACVD) sample after 12 weeks of the intra-abdominal implantation.

	Reference			PACVD-Modified Knitted Implant		
	Adhesion Quality	Adhesion Area	Mode of the Implant Covering	Adhesion Quality	Adhesion Area	Mode of the Implant Covering
	3	1	3	3	0	3
	3	1	3	3	1	3
	3	3	2	3	0	2
	3	2	2	2	2	2
	2	1	3	2	2	2
**Summary**	14	8	13	13	5	12
**Average**	2.8	1.6	2.6	2.6	1.0	2.4
** *S_a_* _1_ **	−0.6					
**Importance**	1	2	1	1	2	1
**Weighted average**	1.8			1.5		
** *S_a_* _2_ **	−0.3					

**Table 8 materials-15-02671-t008:** Evaluation score used to estimate the general tissue response of the tested knitted implant (PACVD modified) that were implanted in the abdominal cavity (in sublay position).

Time of the Implantation (Week)	Score ^1^	Evaluation
2	1.79	No irritation
4	−3.33	No irritation
12	−2.01	No irritation

^1^ Score < 0: the evaluated sample shows a lower irritation reaction compared with the reference (better biocompatibility).

## Data Availability

The data presented in this study are available upon request from the corresponding author.

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
