# Peer review of "Design of New Concept of Knitted Hernia Implant"

_materials, 2022, doi:10.3390/ma15072671_

Round 1
Reviewer 1 Report
Dear Authors,
Thanks for your contribution. Please refer to my suggestion/comments carefully.
The manuscript will be very interesting for audience/reader due to its well written and explanation. However, I would like to suggest and want to get response on few points as below:
- In introduction, please avoid citation that you have cited after a long paragraph like [1-6] and [3,7-15]. Instead of this, please cite in each statement with most updated single reference. Check throughout your manuscript carefully.
- I would like to suggest doing Acute Oral toxicity test, followed by the OECD guidelines.
- Functional moiety determination and physiochemical changes, I would like to suggest doing FTIR analysis of sample.
- What is the main limitation of your present study that may suggest conducting further?
Author Response
Dear Reviewer,
Thank you for your valuable comments, please find the responses below.
Document No. 62/2010 approved by the Institutional Review Board of The Ist Local Ethical Committee For Animal Experiments in Wrocław (Poland) concerns the correction of the number of guinea pigs for irritation studies. We do not publish these studies in this manuscript and we only intend to do so in next publication.
Local Effect After Implantation of the Low-Temperature Plasma-Assisted Chemical-Vapor Deposition Modified Knitted Hernia Implant
Review 1:
Comments and Suggestions for Authors
Dear Authors,
Thanks for your contribution. Please refer to my suggestion/comments carefully.
The manuscript will be very interesting for audience/reader due to its well written and explanation. However, I would like to suggest and want to get response on few points as below:
In introduction, please avoid citation that you have cited after a long paragraph like [1-6] and [3,7-15]. Instead of this, please cite in each statement with most updated single reference. Check throughout your manuscript carefully.
The first citations considered only for one sentence:
The hernia repair, especially by laparoscopy performed, are the most often performed procedures in general surgery [1-6].
The above sentence has been separated. Similar citations for long sentence have been separated.
I would like to suggest doing Acute Oral toxicity test, followed by the OECD guidelines.
The other biocompatibility tests, incl. acute and chronic toxicity, genotoxicity, allergenicity, subdermal irritation, pyrogenicity have been performed acc. suitable part of ISO 10993 standard. The results of above studies will be the subject of next publications
Functional moiety determination and physiochemical changes, I would like to suggest doing FTIR analysis of sample.
The pervious results of the studies, incl. FTIR analysis were implemented at the beginning of Result and Discussion section:
In the ATR – FTIR spectrum of PACVD modified implants, the presence of absorption bands at a wavenumber of 1339 cm-1 associated with the presence of a modified surface (C-F bond) at the stage of low-temperature plasma treatment was found. Process of the sterilization did not effect on the presence of above-mentioned absorption band. A significant increase in the intensity of the absorption bands at l = 2870 cm-1, l = 2920 cm-1 (with the absorption band shift to a higher wavenumber at l = 2917 cm-1) and l = 2950 cm-1 as compared with the unmodified implant ATR-FTIR spectrum [34].
A quantitative EDS microanalysis of the fluorine content on the implant surface directly after the PACVD modification showed a value of 5.22%. This value was slightly reduced after the steam sterilization process to 4.98%. The degree of crystallinity of the sterile implant determined by the WAXS technique was 58.8% [34].
The preliminary study - verification of the degree of implant adhesion was performed taking into account the mechanical test (adhesion strength), SEM observation of the explanted implants and changes in the crystallinity index as well as FTIR verification of the ex-planted implant after primary purification from the tissue [29]. The conducted tests (semi-quantitative evaluation of adhesion according to the developed evaluation criteria) showed that the PACVD modified knitted implants adhered on a significantly smaller surface than in the case of the control sampling. Similar phenomenon was also validated in quantitative assessment of adhesion ex-vivo [29].
The effect of implantation (2, 4 or 12 weeks) on the structural properties of the developed implants with respect to a unmodified sample, the qualitative adhesion degree of internal organs to the modified surface (by SEM), as well as the overgrowth degree by the connective tissue of the unmodified surface with loops was also validated [29].
The statistically insignificant changes in crystallinity index during the intra-peritoneal implantation of the PACVD up to 12 weeks were observed. The ATR-FTIR tests confirmed the presence of the fluoropolymer layers (presence of absorption band at l = 1339 cm-1 corresponding to C-F bond) even after 12 weeks of the implantation [29].
These studies ex-vivo confirmed the better integration of the PACVD modified implant into connective tissue, the stability of the implemented fluoropolymer layer and the reduction of the risk of adhesions with internal organs.
- Kostanek K., Struszczyk M.H., Domagała W., Krucińska I., SURFACE MODIFICATION OF THE IMPLANTABLE KNITTED STRUCTURES FOR POTENTIAL APPLICATION IN LAPAROSCOPIC HERNIA TREATMENTS, Proceedings of FiberMed11, 2011, Tampere, Finlandia, ISBN 978-952-15-2607-7, electronic publishing
What is the main limitation of your present study that may suggest conducting further?
We performed several tests, incl. accelerated aging studies, susceptibility of the PACVD modification for the resterilization process, whole biocompatibility tests required by ISO 10993 standards, explanation investigation of the PACVD stability, biomechanical studies, etc. The results of above research conformed the full performance and safety of developed implants. The last step of the general research is clinical study.
Reviewer 2 Report
This paper was well-organized, the modified knitted hernia implant was fabricated, and many in vivo experiments data were provided to confirm its biomedical application. Many thanks for the valuable work. I suggest this paper should be transferred to a Medical Journal to further confirm its clinic values.
It is interesting, however, if for Materials Journal, the design and related results were limited. It is an original work. More valuable in vivo data was provided compared with other published materials. The paper is well written and organized. The text is clear and easy to read.The conclusions are consistent with the evidence and arguments presented and the authors addressed the main question posed. The quality of figures should be improved, such as the scale bar should be added.
Author Response
Dear Reviewer,
Thank you for your valuable comments. Please find the responses below.
Document No. 62/2010 approved by the Institutional Review Board of The Ist Local Ethical Committee For Animal Experiments in Wrocław (Poland) concerns the correction of the number of guinea pigs for irritation studies. We do not publish these studies in this manuscript and we only intend to do so in next publication.
Local Effect After Implantation of the Low-Temperature Plasma-Assisted Chemical-Vapor Deposition Modified Knitted Hernia Implant
Review 2:
Comments and Suggestions for Authors
This paper was well-organized, the modified knitted hernia implant was fabricated, and many in vivo experiments data were provided to confirm its biomedical application. Many thanks for the valuable work. I suggest this paper should be transferred to a Medical Journal to further confirm its clinic values.
It is interesting, however, if for Materials Journal, the design and related results were limited. It is an original work. More valuable in vivo data was provided compared with other published materials. The paper is well written and organized. The text is clear and easy to read.
The conclusions are consistent with the evidence and arguments presented and the authors addressed the main question posed.
The quality of figures should be improved, such as the scale
The quality of Figure 1b has been corrected for better visibility of the scale. The quality of other figures have been improved.
Reviewer 3 Report
I read the paper about Local Effect After Implantation of the Low-Temperature Plasma-Assisted Chemical-Vapor Deposition Modified Knitted Hernia Implant and found it very poor.
The title is too confusing and one will unable to understand the aim of the work.
The abstract seems unscientific and did not not supported with statistical/mathematical data.
Similarly rest of the parts are poorly presented and I can not recommend it for publication in the current form.
Author Response
Dear Reviewer,
Thank you for your comments. Please find the responses below.
Document No. 62/2010 approved by the Institutional Review Board of The Ist Local Ethical Committee For Animal Experiments in Wrocław (Poland) concerns the correction of the number of guinea pigs for irritation studies. We do not publish these studies in this manuscript and we only intend to do so in next publication.
Local Effect After Implantation of the Low-Temperature Plasma-Assisted Chemical-Vapor Deposition Modified Knitted Hernia Implant
Review 3:
Comments and Suggestions for Authors
I read the paper about Local Effect After Implantation of the Low-Temperature Plasma-Assisted Chemical-Vapor Deposition Modified Knitted Hernia Implant and found it very poor.
The title is too confusing and one will unable to understand the aim of the work.
We cannot agree with the above statement. The title contains both the aspect of the study and the subject of the study, which allows you to precisely determine its purpose.
The abstract seems unscientific and did not not supported with statistical/mathematical data.
The summary has been correctly compiled – statistical data, due to the context, including the ethical aspect, cannot be taken into account. The research was based on acceptable ethical and normative requirements.
Similarly rest of the parts are poorly presented and I can not recommend it for publication in the current form.
The above statement is unfair – the research program, the way of presenting the results and their discussion has been developed correctly and meets qualitatively scientific standards.
Reviewer 4 Report
Comments
The authors evaluated the biocompatibility including irritation, adhesion and local tissue response after the implantation of monolayer fluoropolymer modified knitted implants by PACVD. Additionally, different scoring systems were established to evaluate the implant and reference quantitatively. This study was based on their previous report (Ref. 29) and the superiority of the fluoropolymer modified implant was demonstrated. This is a systematic study to evaluate the biosafety of the fluoropolymer modified implant. I recommended the publication of this manuscript in Materials after addressing the following issues.
- Though the characterization of the monolayer fluoropolymer implant was reported in their previous study, the characterizations of the implant such as SEM, XPS, FTIR results should be briefly described in the manuscript. Besides, the summary of the characterization of the monolayer of fluoropolymer e. g. Line 471-475, should be removed to the beginning of results and discussion.
- The meaning of the score in the abstract should be briefly explained right after the score, which is helpful for better understanding.
- The abbreviation of PACVD should be appeared at the first place, and then only abbreviation used in the following text. Besides, the full name of PACVD is different in the abstract and the main text, which should be unified.
- The quality of Figure 1b is poor. And the scale bar of the figures should be provided, even digital photographs.
- The PACVD modified implants were reported in the authors previous study (Ref. 29), which was published in 2013. Therefore, I suppose that the use of "newly developed knitted" in this manuscript is improper.
Author Response
Dear Reviewer,
Thank you for your valuable comments. Please find the responses below.
Document No. 62/2010 approved by the Institutional Review Board of The Ist Local Ethical Committee For Animal Experiments in Wrocław (Poland) concerns the correction of the number of guinea pigs for irritation studies. We do not publish these studies in this manuscript and we only intend to do so in next publication.
Local Effect After Implantation of the Low-Temperature Plasma-Assisted Chemical-Vapor Deposition Modified Knitted Hernia Implant
Review 4:
Comments and Suggestions for Authors
Comments
The authors evaluated the biocompatibility including irritation, adhesion and local tissue response after the implantation of monolayer fluoropolymer modified knitted implants by PACVD. Additionally, different scoring systems were established to evaluate the implant and reference quantitatively. This study was based on their previous report (Ref. 29) and the superiority of the fluoropolymer modified implant was demonstrated. This is a systematic study to evaluate the biosafety of the fluoropolymer modified implant. I recommended the publication of this manuscript in Materials after addressing the following issues.
Though the characterization of the monolayer fluoropolymer implant was reported in their previous study, the characterizations of the implant such as SEM, XPS, FTIR results should be briefly described in the manuscript. Besides, the summary of the characterization of the monolayer of fluoropolymer e. g. Line 471-475, should be removed to the beginning of results and discussion.
The suggested sentence was replaced to the beginning of Result and Discussion section:
In the ATR – FTIR spectrum of PACVD modified implants, the presence of absorption bands at a wavenumber of 1339 cm-1 associated with the presence of a modified surface (C-F bond) at the stage of low-temperature plasma treatment was found. Process of the sterilization did not effect on the presence of above-mentioned absorption band. A significant increase in the intensity of the absorption bands at l = 2870 cm-1, l = 2920 cm-1 (with the absorption band shift to a higher wavenumber at l = 2917 cm-1) and l = 2950 cm-1 as compared with the unmodified implant ATR-FTIR spectrum [34].
A quantitative EDS microanalysis of the fluorine content on the implant surface directly after the PACVD modification showed a value of 5.22%. This value was slightly reduced after the steam sterilization process to 4.98%. The degree of crystallinity of the sterile implant determined by the WAXS technique was 58.8% [34].
The preliminary study - verification of the degree of implant adhesion was performed taking into account the mechanical test (adhesion strength), SEM observation of the explanted implants and changes in the crystallinity index as well as FTIR verification of the ex-planted implant after primary purification from the tissue [29]. The conducted tests (semi-quantitative evaluation of adhesion according to the developed evaluation criteria) showed that the PACVD modified knitted implants adhered on a significantly smaller surface than in the case of the control sampling. Similar phenomenon was also validated in quantitative assessment of adhesion ex-vivo [29].
The effect of implantation (2, 4 or 12 weeks) on the structural properties of the developed implants with respect to a unmodified sample, the qualitative adhesion degree of internal organs to the modified surface (by SEM), as well as the overgrowth degree by the connective tissue of the unmodified surface with loops was also validated [29].
The statistically insignificant changes in crystallinity index during the intra-peritoneal implantation of the PACVD up to 12 weeks were observed. The ATR-FTIR tests confirmed the presence of the fluoropolymer layers (presence of absorption band at l = 1339 cm-1 corresponding to C-F bond) even after 12 weeks of the implantation [29].
These studies ex-vivo confirmed the better integration of the PACVD modified implant into connective tissue, the stability of the implemented fluoropolymer layer and the reduction of the risk of adhesions with internal organs.
- Kostanek K., Struszczyk M.H., Domagała W., Krucińska I., SURFACE MODIFICATION OF THE IMPLANTABLE KNITTED STRUCTURES FOR POTENTIAL APPLICATION IN LAPAROSCOPIC HERNIA TREATMENTS, Proceedings of FiberMed11, 2011, Tampere, Finlandia, ISBN 978-952-15-2607-7, electronic publishing
The meaning of the score in the abstract should be briefly explained right after the score, which is helpful for better understanding.
The below-listed sentence was added to abstract:
The ultimate aspect of biological evaluation is biofunctionality and evaluation of the local tissue response after implantation, that results in the determination of the whole aspects of local biocompatibility of the implemented synthetic material The implantation of the PACVD modified materials in muscle allows to estimate of the local irritation effect on the connective tissue with the determination of the risk of the scar formation, whereas implantation of the above-mentioned knitted fabric in abdominal wall evaluate the risk of the fistula formation – the main post-surgical complications […].
[…] The comparative semi-quantitative histological assessment, included the severity of inflammatory cells (multinucleated cells, lymphocytes, plasma cells, macrophages, giant cells) and the tissue response (necrosis, neovascularization, fibrosis and fat infiltration) on a five-point scale.
The abbreviation of PACVD should be appeared at the first place, and then only abbreviation used in the following text. Besides, the full name of PACVD is different in the abstract and the main text, which should be unified.
The above aspects have been corrected.
The quality of Figure 1b is poor. And the scale bar of the figures should be provided, even digital photographs.
The Figure 1b has been corrected.
The PACVD modified implants were reported in the authors previous study (Ref. 29), which was published in 2013. Therefore, I suppose that the use of "newly developed knitted" in this manuscript is improper.
The sentence has been corrected.
Round 2
Reviewer 1 Report
Thanks for your kind responses.
Author Response
Dear Reviewer,
thank you for your effort and acceptance.
Reviewer 2 Report
I suggest the acceptance in current form.
Author Response

(The authors gave the same response as above.)

Reviewer 3 Report
I read the revised MS and found that there no or little revision has been carried out.
The authors replied to my previous comments and responded with UNFAIR replies.
The title still lacks comprehensiveness and abstract lacks statistical significance differences.
In introduction the authors just added some reference numbers and no new sentence.
Author Response
Dear Rewiever,
Thank you for your comments and effort.
The title Evaluation of the Local Effect After Implantation of the Low-Temperature Plasma-Assisted Chemical-Vapor Deposition Modified Knitted Hernia Implant covers both: subject of the study as well as range of the investigation. The local effect after implantation is a standard term in biocompatibility studies determines the whole aspect of the implantation effect estimation in local range. The above term is normally used to describe the scope of biocompatibility testing and is used globally due to its comprehensiveness. Taking into account above we postulate to leave the title in proposed by us form.
The statistical significance differences is not possible to introduce due to the methodology acc. ISO 10993-6 applied with limited numbers of animals resulted from the ISO 10993-2. Moreover, for the estimation of the risk of the fistula formation the semi-quantitative analysis has been adopted.
We hope our response and statement is now sufficient and clear.
Best regards - Authors
This manuscript is a resubmission of an earlier submission. The following is a list of the peer review reports and author responses from that submission.